# Effects of Maternal Pre-Pregnancy BMI on Preterm Infant Microbiome and Fecal Fermentation Profile—A Preliminary Cohort Study

**DOI:** 10.3390/nu17060987

**Published:** 2025-03-11

**Authors:** Kristy L. Thomas, Amy E. Wahlquist, Dalton James, William Andrew Clark, Carol L. Wagner

**Affiliations:** 1Department of Biomedical Sciences, Quillen College of Medicine, East Tennessee State University, Johnson City, TN 37614, USA; thomkris@musc.edu (K.L.T.); clarkw@etsu.edu (W.A.C.); 2Department of Rehabilitative Sciences, College of Health Sciences, East Tennessee State University, Johnson City, TN 37614, USA; 3Center for Rural Health and Research, Department of Biostatistics and Epidemiology, College of Public Health, East Tennessee State University, Johnson City, TN 37614, USA; 4Department of Biological Sciences, College of Arts and Sciences, East Tennessee State University, Johnson City, TN 37614, USA; 5Division of Neonatology, Department of Pediatrics, Shawn Jenkins Children’s Hospital, Medical University of South Carolina, 10 McClennan Banks Drive, MSC 915, Charleston, SC 29425, USA

**Keywords:** preterm neonate, preterm infant, body mass index, microbiome, fecal, gut, short chain fatty acids, SCFA, maternal, developmental programming

## Abstract

**Objective:** This feasibility, proof-of-concept study aimed to assess the impact of maternal pre-pregnancy body mass index (BMI) on preterm infant fecal fermentation and microbiome. **Study Design:** An infant cohort study (n = 54) in the NICU at MUSC from June 2021 to September 2022 was grouped according to maternal pre-pregnancy BMI—normal weight (<25 kg/m^2^), overweight (25–29.9 kg/m^2^), and obese (≥30 kg/m^2^). All fecal samples were subjected to 16s rRNA isolation and analysis, as well as short chain fatty acid (SCFA) extraction and analysis. **Results:** Preterm infants born to overweight and obese mothers did not have differences in microbial diversities but did have different bacterial taxonomic composition and lower relative abundance levels of taxa than those born to normal-weight mothers. While controlling for covariates, we found SCFA propionic acid to be higher and more significant in infant stools born to mothers with a higher pre-pregnancy BMI. **Conclusions:** This is a novel study investigating the microbiome and SCFA in premature infants while considering maternal pre-pregnancy BMI. This study adds to the current literature, in that the preterm infant gut is generally lower in microbial diversity which can impact infant health. Thus, it is important to understand the mechanisms necessary to modulate the microbiome of preterm infants to improve their health outcomes.

## 1. Introduction

Preterm birth is a significant public health concern with high morbidity and mortality rates globally, including in the United States (US) [1,2]. South Carolina (SC) has one of the highest rates of prematurity, especially among the Black population [1]. Preterm infants are more susceptible to adverse health outcomes due to prolonged hospital stays, including respiratory issues, nutritional deficiencies and extrauterine growth failure, neurological complications, necrotizing enterocolitis (NEC), and bronchopulmonary dysplasia (BPD), among others [3,4,5,6,7].

According to the March of Dimes Maternity Care Desert Report, many pregnant women in South Carolina have limited access to healthcare and live in what is viewed as a maternity care desert [8]. Surgo Ventures has rated South Carolina as having a very high rating on the maternal vulnerability index (92/100), which correlates to the maternal mortality rate [8,9]. Maternal health continues to impact adverse events during pregnancy, including premature birth and higher rates of maternal perinatal deaths, which are also complicated by higher rates of type 2 diabetes, chronic hypertension, and obesity [8,9,10,11]. The leading cause of death in neonates is preterm birth [12], accounting for 35% of neonatal deaths [13]. According to the Centers for Disease Control and Prevention (CDC), in 2021, the preterm birth rate in the US was 10.5% overall and 12.1% in SC [2]. In the US, populations of Black mothers have disproportionally higher rates of preterm births than mothers of other races at 14.4% [14,15].

Preterm births increase the risk of morbidity and mortality for infants after birth. The mother’s milk (MoM) is the most beneficial milk given to preterm infants to decrease infection, morbidity, and mortality [16], decrease intestinal permeability, and increase gut maturation [17]. Both maternal and donor breast milk have been shown to reduce the risk of necrotizing enterocolitis (NEC) [17,18,19,20,21]. MoM or donor breast milk is considered the best feeding option for infants in the neonatal intensive care unit (NICU) [22,23]. The “developmental origins of health and disease” (DOHaD) theory suggests that health outcomes are shaped during the first 1000 days of development (two months prior to conception up to two years of life), highlighting the importance of studying the relationships among maternal nutritional status, breast milk components, and the health outcomes of preterm infants in later life, which have been explored to a limited extent [24,25,26,27,28].

Preterm infants have different gut bacteria than full-term healthy infants [29]. Premature neonates nutritionally require fortified HBM for high protein, calcium, and phosphate intake to prevent extrauterine growth restriction (EUGR), as unfortified HBM is not sufficient for optimal growth and body composition changes [21,23]. Preterm birth and other factors including advanced maternal age, maternal illness, primiparity, elevated body mass index (BMI), gestational diabetes, and cesarean delivery contribute to low rates of exclusive breast feedings [23,30,31]. The low rate of exclusive maternal breast feeding warrants an increased use of donor breast milk, which has been associated with decreased NEC in preterm infants when compared to those fed with formula [32,33]. The microbiome of all infants—both term and preterm—changes rapidly after parturition, affected by maternal skin-to-skin contact, oral transfer, and human breast milk (HBM) microbiomes via transmission from the mother to the infant [34], which impacts the overall health of the infant.

The interplays between maternal obesity and other health disturbances, such as hypertension, pre-eclampsia, and dyslipidemia, on maternal nutrition and infant outcomes, is poorly understood. The prevalence of obesity is increasing in the US, but there is currently no universal prevention method [35]. Animal models show that the offspring of obese mothers are more susceptible to obesity and that the transfer of obesity from mother to offspring can occur through mammalian milk [36]. Maternal factors, including pre-pregnancy BMI, maternal health, and nutrition status, influence breast milk components and microbiome [15,37,38]. Maternal obesity is associated with reduced duration and exclusivity of breastfeeding, necessitating the use of donor breast milk in preterm infants [23,30,33]. Maternal nutrition impacts the growth and development of the fetus in utero and later in life.

While any associations between human milk components and subsequent child adiposity or obesity in preterm infants is debated [39], some propose that the milk composition of obese mothers could contribute to, or could be less protective against, child obesity through the alteration of components that influence satiety, growth, inflammatory factors, or carbohydrate–microbial interactions [40]. The complicating factor of preterm birth and MoM vs. donor breast milk (DBrM) dose on these effects remains unknown.

## 2. Objectives or Purpose

This study aimed to address gaps in the understanding of how maternal pre-pregnancy BMI impacts preterm infant fecal fermentation and microbiome profiles, which may be indicative in future health and development outcomes. We conducted a cross-sectional study of preterm infants to explore how maternal anthropometric status could correspond to the development and maturation of preterm infant microbiome and fecal fermentation profiles from a diverse patient population.

**Hypothesis** **1.***Maternal BMI is related to (a) HBM SCFA composition, (b) HBM microbiome, and to c) her recipient’s (the premature infant) microbiome and fecal fermentation profile, through enteral breast milk feedings, resulting in less diverse profiles in preterm infants born to mothers with higher BMIs*.

**Hypothesis** **2.***Factors such as gestational age at birth, days postpartum, race/ethnicity, delivery mode, MoM dose, and infant feeding type would correlate to the preterm infant microbiome and fecal fermentation profiles*.

## 3. Materials and Methods

### 3.1. Study Design and Subjects

This study was a preliminary cohort study to determine feasibility for a larger study of preterm infants (n = 54) at ≤ 37 weeks of gestation, which were admitted to Shawn Jenkins Children’s Hospital’s (SJCH) Level IV Neonatal Intensive Care Unit (NICU) between May 2022 and September 2022 in South Carolina, to assess the impact of maternal pre-pregnancy BMI on infant fecal fermentation and microbiome patterns. Infants were categorized into maternal pre-pregnancy BMI groups using the following standardized groupings according to the World Health Organization (WHO) [41]: (1) normal weight 18.5 to 24.9 (kg/m^2^), (2) overweight 25 to 29.9 (kg/m^2^), and (3) obese ≥ 30 (kg/m^2^), based on pre-pregnancy weight. Infants were excluded from the study if they had been on antibiotics. This study was approved by the Institutional Review Board at MUSC and deemed exempt (MUSC IRB exempt PRO00120615). Deidentified demographic and pertinent medical information about the study subjects was collected from the electronic medical records and entered into the REDCap Database (Table 1 and Table 2).

### 3.2. Stool Collection and Microbiome and Short Chain Fatty Acid Analysis

Stool samples were collected during the general clinical hands-on time for each infant. The samples were placed in a sterile container and immediately stored at −80 °C before being sent as deidentified samples to East Tennessee State University (ETSU) for sample analysis.

Fecal genomic DNA was extracted from samples according to manufacture specifications using the Qiagen DNeasy PowerSoil Pro Kit DNA isolation kit, (Qiagen, Hilden, Germany) (Catalog No. 47014). DNA from the fecal samples were quantified via ThermoScientific™ NanoDrop™ One Microvolume UV-Vis Spectrophotometer (Thermo Fisher Scientific, Waltham, MA, USA) to standardize the quantity of DNA for DNA analysis.

Using the Klindworth et al. [42] method, the V4 region of the 16S rRNA gene was amplified from the extracted DNA using primers 341f and 785r and modified with adapters for Illumina MiSeq sequencing. DNA was fragmented and tagged with a 615f/806r adapted sequence before polymerase chain reaction amplification. The initial PCR consisted of 2 × KAPA HiFi HotStart ReadyMix Taq (Roche, Indianapolis, IN, USA), 1.5 μM each primer, and 2.5 μL DNA; then, 20 µL of AMPure XP beads (Agencourt, Beverly, MA, USA) was used and the sample was eluted in 50 µL of Tris-HCl. Nextera XT indexes Illumina (San Diego, CA USA). The standard sensitivity kit confirmed the final product sizes and concentrations on an Agilent Bioanalyzer (Santa Clara, CA, USA). The final library was then diluted to 5 pM, combined with 20% of a 10 pM PhiX library control, and then paired-end sequencing of 275 bp was performed using a v3 (600 cycles) flow cell of an Illumina MiSeq sequencer at the University of Tennessee Genomics Core.

Short-chain fatty acid (SCFA) extractions were performed using a modified procedure developed by Schwiertz et al. [43]. Samples were placed in a 2 mL amber auto-injection vial and stored at −80 °C until being analyzed using a Shimadzu GC2010 gas chromatograph equipped with a Phenomenex Zebron ZB-Wax Plus capillary column (part # 7HG-G013-11). Samples were run in duplicate, and the values for each participant were averaged.

### 3.3. Metagenomic Analysis

Operational Taxonomic Unit (OTU) clustering and taxonomic analyses were performed using CLC Genomics Workbench v. 23 and CLC Microbial Genomics Module v. 2.5 (Qiagen, Hilden, Germany) as previously described [44].

Alpha diversity measures were analyzed via the Kruskal–Wallis test to determine whether there were significant differences between the BMI groups, and the Mann–Whitney test was then used for multiple pair-wise comparisons to determine which specific groups followed different distributions. We used the Shannon index to calculate microbial diversity, which accounts for bacterial richness and evenness. The Shannon index is more sensitive to rare species relative to species richness than Simpson’s Index, which is more sensitive to dominant species [45].

Beta diversity measures were performed with Bray–Curtis (measure of dissimilarity) and Jaccard (similarity and dissimilarity) measurements. A permutational multivariate analysis of variance (PERMANOVA) test was applied to the beta diversity measures to detect differences between the groups with the level of accepted statistical significance set at *p* < 0.05.

To further assess the similarity or dissimilarity between the preterm groups, we determined which OTUs had the most significantly different abundances across all the samples by utilizing a generalized linear model (GLM) differential abundance test after performing TMM (trimmed mean of M-values) normalization on all the samples with the CLC Genomic Workbench Tool for differential abundance analysis.

### 3.4. Statistical Analysis

This study was designed as a feasibility, proof-of-concept study. The initial sample size and power calculations indicated that a study size of at least 200 subjects would be needed to have sufficient power to detect the expected differences in microbiome outcomes between the BMI categories. Simple descriptive statistics, including mean, standard deviation, minimums, maximums, frequencies, and percentages, as appropriate, were used to summarize both the demographic and clinical variables overall for the total cohort (Table 1) and by the BMI group (Table 2). The BMI groups were compared via ANOVA models (continuous outcomes) or Chi-squared/Fisher’s Exact tests (categorical outcomes), as appropriate.

Bivariate associations between the demographic or clinical characteristics across all samples and the outcomes from the SCFA were performed using *t*-tests (variables with two groups such as c-section [yes/no]) or one-way (ANOVA) models (variables with three or more groups such as pre-pregnancy BMI), as appropriate. Multivariable linear regression models were then used to examine the association between each outcome of interest from the SCFA analysis (acetic acid, butyric acid, propionic acid, the total of acetic + butyric + propionic acid) and the alpha diversity measure of Shannon Entropy or the Shannon diversity index for pre-pregnancy BMI group (normal vs. overweight vs. obese) in the context of other covariates of interest (race/ethnicity, c-section, day of life, gestational age, and MoM dose or percent of MoM provided). The MoM dose provided was calculated based on the total mL of MoM plus the donor milk at the time of sample collection. All analyses were considered statistically significant at an α = 0.05 two-sided level of significance. The analyses were generated using SAS software, Version 9.4 of the SAS System for Windows.

The primary outcomes of this study included the following: (1) stool microbiome profile and (2) fecal fermentation profile (FFP) inclusive of acetic, propionic, isobutyric, butyric, isovaleric, valeric, isocaproic, caproic, heptanoic, octanoic, and acetic, propionic, and butyric acids combined. The resulting microbiome and FFP profiles were utilized to compare preterm infants born to mothers of differing pre-pregnancy BMI (normal vs. overweight vs. obese) to determine if there were any significant differences between groups.

## 4. Results

### 4.1. Subject Demographics

This study consisted of a cohort study of 54 preterm infants (admitted to the Level IV Neonatal Intensive Care Unit at MUSC in South Carolina) to assess the impact of maternal pre-pregnancy BMI on infant fecal fermentation and microbiome patterns. The study demographics (Table 1) of the preterm infants included gestational ages ranging from 24 to <37 weeks and birth weights of 620 g to 2350 g. The racial composition of the study population was diverse. We noted an uneven sample distribution of BMI among the Black mothers, with normal-weight mothers accounting for 12% (n = 3) and obese mothers (n = 22) accounting for 88% of all Black mothers in the study sample. More infants were born by cesarean section than vaginally. Fecal sample collection ranged from day 2 to 95 of life. Most of the preterm infants in the study consumed a combination of mother’s milk and donor milk, where only 9% consumed mother’s milk only and 9% consumed donor milk only. None of the preterm infants were formula fed.

The preterm infant subjects were stratified by the birth mothers’ pre-pregnancy BMI categories (Table 2). Maternal pre-pregnancy BMI ranged from 18 kg/m^2^ to 46.7 kg/m^2^, resulting in most infants being categorized into the maternal BMI groups of overweight (27%) or obese (54%).

### 4.2. Metagenomic Analysis Results

We compared the relative abundances of the microbiome profiles from premature neonates born to mothers of differing pre-pregnancy BMIs. After quality filtering and length trimming, 3,888,004 16S rRNA sequences were analyzed, with an average number of taxonomically assigned, high-quality sequences of 72,000 per sample. The taxonomic assignment of the sequences showed that the composition of the premature neonate fecal samples at the phylum level was dominated by Proteobacteria at 51.6%, followed by Firmicutes at 35.1% (Figure 1a). Proteobacteria (multiple morphologies) and Firmicutes (bacilli) were also the two predominant taxa in all the BMI groups, with Proteobacteria making up 47% of the normal-weight group, 60% in the overweight group, and 49% in the obese group and Firmicutes making up 37%, 26%, and 39%, respectively (shown in Figure 1a). In the fecal samples, the most common genera were *Enterococcus* (lactic acid bacteria from the order of Lactobacillales) at 8.95% and *Klebsiella* (from the order of Enterobacteriales) at 8.90%, followed by *Bacteroides* (Bacteroidales) at 7.89%, *Veillonella* (Clostridiales) at 5.63%, and *Streptococcus* (Lactobacillales) at 3.70%. The normal-weight group infants had *Enterococcus* at 12.85%, *Bacteroides* at 10.70%, *Enterobacter* (Enterobacteriales) at 10.22%, *Veillonella* at 8.57%, and *Citrobacter* (Enterobacteriales) at 8.26% as the top five most common genera. The overweight maternal BMI group infants had *Veillonella* at 8.46%, *Klebsiella* at 8.40%, *Bacteroides* 5.80%, *Enterococcus* at 5.35%, and *Bifidobacterium* (Bifidobacteriales) at 4.17%, while the obese maternal BMI group infants had *Klebsiella* at 10.39%, *Enterococcus* at 9.40%, *Bacteroides* at 7.96%, *Streptococcus* at 5.29%, and *Veillonella* at 3.21% as the top five genera (Figure 1b). We also compared the relative abundances of the microbiome profiles from premature neonates born to mothers of differing BMI at the class, order, family, and species levels (data not reported) (Appendix A).

There were no statistically significant differences in the alpha diversity metrics of the fecal samples from the preterm infants that were grouped according to maternal pre-pregnancy BMI. The Shannon index was not significant between the maternal pre-pregnancy BMI groups (Kruskal–Wallis test: *p* = 0.5 for all groups; Mann–Whitney test: *p* = 0.7 for normal v. obese; *p* = 0.3 for normal v. overweight; and *p* = 0.3 for obese v. overweight) (Figure 2a). Simpson’s Index was not significant between the groups (Kruskal–Wallis test: *p* = 0.5 for all groups. Mann–Whitney test: *p* = 0.6 for normal v. obese; *p* = 0.2 for normal v. overweight; and *p* = 0.4 for obese v. overweight) (Figure 2b).

Figure 3a,b display the 3-dimensional Principal Coordinate of Analysis plots showing each sample as a colored sphere, with the obese group in red (n = 29), overweight group in green (n = 15), and normal-weight group in purple (n = 10). The lack of sample group clustering between the groups of samples indicates no significant differences in the beta diversity for Bray–Curtis (*p* = 0.32) or Jaccard (*p* = 0.41) in the microbiomes of the premature neonates born to mothers with differing BMIs.

For the following differential abundance results, the family, genus, and species are denoted as f_x, g_x, or s_x. Proteobacteria (f_*Enterobacteriaceae*), Bacteroidetes (f_*Rikenellaceae*, g_unknown, f_*Porphyromonadaceae*, g_*Parabacteroides*, f_*Bacteroidaceae*, g_*Bacteroides*), Firmicutes (f_*Veillonellaceae*, g_*Veillonella*, s_*dispar*), and Proteobacteria (f_*Enterobacteriaceae*, g_*Klebsiella*, s_*oxytoca*) were all significantly different in the fecal samples of the preterm infants born to mothers of normal weight and had higher relative abundances than the infants born to overweight or obese mothers (Figure 4). Firmicutes (f_[*Tissierellaceae*], g_*Peptoniphilus*), Bacteroidetes (f_*Bacteroidaceae*, g_*Bacteroides*, s_*caccae*), and Proteobacteria (f_*Enterobacteriaceae*, g_unknown) had higher relative abundances and were significantly different in the fecal samples of the preterm infants born to overweight mothers than the preterm infants born to normal-weight or obese mothers (Figure 2). Surprisingly, even though the preterm infants born to mothers who were obese had significantly different abundances in taxa than the infants born to normal-weight and overweight mothers, they had low relative abundance levels of bacterial taxa compared to the infants born to mothers of normal or overweight BMIs (Figure 2).

### 4.3. Fecal Fermentation Profile (FFP)

We observed significant (*p* = 0.020) and increased propionic acid concentrations in the preterm neonate stools of infants born to mothers with higher pre-pregnancy BMI while controlling for other factors that may impact microbiome and SCFA profiles (race/ethnicity, delivery mode, infant day of life, infant gestational age, and dose of MoM) (Table 3). A similar increase but not significant (*p* = 0.050) trend in acetic acid was observed in preterm neonates born to mothers with a higher pre-pregnancy BMI. No significant differences were observed in other SCFA concentrations or APB (acetate, propionate, and butyrate) combinations in infants born to mothers of differing BMIs.

### 4.4. Other Factors That May Impact Microbiome

Factors that may impact preterm infant fecal microbiome include the infant’s day of life, infant gestational age, birth mode, race and ethnicity, and infant feeding type (MoM dose). We conducted a microbiome analysis for each of these factors to determine if the results we observed could be due to other factors that may impact the infant’s fecal microbiome profiles. We observed no significant differences in the alpha diversity when examining all the groups in the context of each maternal/infant factor using an ANOVA-like test (Kruskal–Wallis) (Table 4). There were significant differences observed, however, between some groups in the context of each maternal/infant factor when using the Mann–Whitney test. Regarding the infants’ weeks of life, we found significant differences between weeks 1 and 3 (*p* = 0.02). The analysis for MoM dose was significant for DBrM only vs. mostly DBrM (*p* = 0.03) but not for MoM vs. any other factor. We found significant differences in the beta diversity measures between the groups in the context of delivery mode for both Bray–Curtis (*p* = 0.02) and Jaccard (*p* = 0.03), as well as a trend towards significance between the groups for gestational age for both Bray–Curtis (*p* > 0.05) and Jaccard (*p* = 0.06) (Table 4).

## 5. Discussion

### Summary of Findings

This study aimed to address the gaps in our understanding of how maternal pre-pregnancy BMI is related to preterm infant fecal fermentation and microbiome profiles, which has indications for the future health and developmental outcomes of infants, by exploring how maternal anthropometric status influences the development and maturation of preterm infants, particularly preterm infant microbiome and fecal fermentation profiles.

When looking at the relative abundance of the top phyla in the preterm infants grouped according to maternal BMI, all the groups had *Enterococcus* and *Klebsiella* as the top two predominant phyla. The normal-weight group infants had *Bacteroidetes* and *Enterobacter*, while overweight group had *Veillonella* and *Klebsiella*, and the obese group had *Klebsiella* and *Enterococcus* as their top two phyla.

We saw no alpha or beta diversity differences between the preterm neonates born to mothers of any BMI status. This may be due to the lack of a mature gut microbiome in preterm infants, as most (81%) of the infants were under five weeks of age. Maturation of the gut microbiome starts shortly after birth, and it does not fully mature until the age of two years [46].

We noted that there were significant differences when looking at the differential abundance of Proteobacteria, Bacteroidetes, and Firmicutes in the fecal samples of the preterm infants born to mothers of a normal weight, which had higher relative abundances than infants born to overweight or obese mothers. In addition, Firmicutes, Bacteroidetes, and Proteobacteria were higher in relative abundance in preterm infants born to overweight mothers and were significantly different in abundance than preterm infants born to normal-weight or obese mothers. Bacteroidetes and Firmicutes produce butyrate and riboflavin (vitamin B2), positively affecting gut health [46,47]. While Bacteroidetes are correlated with a lower body mass index, Firmicutes can interact with the mucosa of the intestines and contribute to host homeostasis [48,49,50]. Proteobacteria aid in metabolism, gut barrier function, and the synthesis of vitamins K and B2 [51]. There have been proposed links among Proteobacteria, dysbiosis, metabolic syndrome, and inflammatory bowel disease [52,53]. Similar to the other studies, we aimed to characterize the preterm infant gut microbiome. We observed no differences in the alpha or beta diversity fecal microbiomes between the groups but did notice differential abundances between the preterm groups. More research is needed to observe the microbial patterns and changes early in infancy to understand the influence of maternal pre-pregnancy BMI on the preterm infant microbiome.

Even though the overweight and obese groups’ infants had lower relative abundance levels of bacterial taxa compared to the infants born to mothers with a normal weight, due to our small sample size, we observed no significant differences. With a more extensive study using more subjects, we would expect to see significant differences. The effect of maternal BMI would likely result in the infants having less diversity in their microbiome.

Short chain fatty acids are the metabolites of gut bacteria fermenting nondigestible fiber and other undigested nutrients in the colon thus reflecting the gut bacterial flora. Short chain fatty analysis performed using a multivariable linear regression model, while controlling for various factors (race/ethnicity, c-section, day of life, gestational age, and percent of MoM’s milk provided), showed significant relationships for propionic acid but not with any other short chain fatty acids or APB combinations in the fecal samples from preterm infants born to mothers of differing BMIs. Propionic acid, predominantly produced by Firmicutes and Bacteroidetes, is a satiety inducer, which also plays a role in cholesterol (lowering LDL and increasing HDL), lipid (modulation of hepatic lipid accumulation), and glucose (gluconeogenic) metabolism [54,55,56,57,58,59]. Studies in children have shown higher concentrations of propionic acid in the feces of overweight children than those of normal weight [60]. We have made similar observations in this study. This may suggest a mechanism to help attenuate the effects of obesity early in the child’s development. Our findings add to the existing literature, as propionate in HBM has not been well studied and little is known about maternal BMI and propionic acid concentrations.

When analyzing the microbiome and fecal fermentation profiles for other factors that may alter the microbiome in preterm infants, we observed significant differences in the alpha diversity for infant weeks of life between weeks 1 and 3, feeding type between MoM only and DBrM only, as well as DBrM only and combination milk, and MoM dose between donor milk and mostly donor milk (*p* = 0.03) but not MoM vs. combination. For the beta diversity, both Bray–Curtis and Jaccard tests were significant for differences in delivery mode. We expected that factors such as gestational age at birth, infant weeks of life, race/ethnicity, delivery mode, MoM dose, and infant feeding type would impact the infant fecal microbial diversity and fecal fermentation profiles. We observed some differences and, if the infants had a longer time since the inoculum of the milk microbiome, we may subsequently have observed significant differences in the infant microbial and fecal fermentation profiles.

Even though this was a small-scale preliminary study on preterm infants, this study had several strengths. Our subjects were a diverse population of preterm infants from Charleston, South Carolina, and the surrounding areas. We had diversity within our subject pool, which is not typically seen in infant or milk studies, with 46% of our subjects being Black. Our subject data and fecal samples were collected before the infants were in an external environment. This allowed us to control for any external factors related to the environment when collecting microbiome samples. The anthropometric and demographic data were standardized and not self-reported. We conducted a novel study, as little is known about preterm infant SCFA profiles and only a few microbiome analyses have been completed within this population.

One limitation of this study was that we did not control for the time of sample collection; however, the timing of sample collection between the groups was not significant (*p* = 0.5). Preterm infants are early in the development of their microbiome. Starting earlier and watching the preterm infant microbiome maturation progress would have been beneficial for this study. The study did not have a detailed analysis of the infant diet (i.e., how much DBrM only or MoM only was given at feeding). Because we used 16s rRNA sequencing, we did not have the comprehensive and complete picture that shotgun sequencing would have provided in our analysis. Even though the study had a fair sample size, it was underpowered for microbiome analysis. While we acknowledge that premature birth comes with late onset lactogenesis, observing more infants which exclusively had MoM would have been beneficial. We did not analyze the MoM pool or the DBrM pool. Preterm infants must receive milk from their mother or a human milk bank to prevent infections and promote growth and development. Our study was a preliminary study and not longitudinal in scope. A longitudinal study would allow for the observation of the infant gut microbiome maturation over time. Despite having a decent number of Black mothers, we had a lack of Black mothers with a normal BMI. We had confounders for the days of life of the infant at sample collection, where the infant age ranged from 2 to 95 days. To adjust for cofounders, we should have limited preterm infant age to 30–45 days old. This is so that we would be able to observe infant microbiome after colonization from the mother and before other microbiome colonization influences. Factors including maternal diet, medication (maternal and infant), antibiotic exposure (maternal and infant), infant feeding initiation, family members in the household (father, siblings, etc.), and pets influence the development of the infant’s microbiome and should be considered in future studies.

## 6. Conclusions

While the data from this cohort study did not show significant differences in the infant microbial diversity due to maternal pre-pregnancy BMI, the SCFA propionic acid was significant (*p* = 0.021) and increased with increasing maternal BMI, while controlling for covariates (race/ethnicity, c-section, infant day of life, gestational age, pre-pregnancy BMI, and % of MoM dose). We saw significant differences when examining the other factors that may impact the preterm infant microbiome. This study found that the infant weeks of life, infant feeding type, and delivery mode contribute to microbiome diversity. With a larger sample size, we might have seen significant differences in the infant weeks of life and gestational age for SCFA and beta diversity, respectively. In this small sample, we reject our hypothesis that maternal pre-pregnancy BMI is correlated with the infant’s microbiome and fecal fermentation profiles. With more refined analyses that can control for additional potential confounders, such as infant days of life/sample collection, and include longitudinal sampling and a more detailed analysis of infant diet (e.g., the dose of mother’s milk vs. donor milk, initiation of fortification), the impact of maternal pre-pregnancy BMI on preterm infant microbiome, fecal fermentation, and health status could be better assessed.

### Suggestions for Future Research

The relationship between maternal pre-pregnancy BMI and premature neonatal fecal SCFA and microbiome profiles needs further study. Future research studies should include a larger, more diverse study population to ensure variability of BMI groups within the racial/ethnic groups. Future studies should also be longitudinal (2 years) to observe the growth and development of the infant and their microbiome over time and have a standardized collection of stool samples at standardized timepoints, consistency in the collection of stool samples from mother and child, and collection of MoM and donor milk to compare all the samples at the same time. Health information, including medications, antibiotics, and mode of ventilation (respirator, nasal cannula, or room air), should be collected to account for variables that may impact the microbiomes and SCFA of both mother and child. Lastly, maternal and fecal blood should be collected and analyzed for inflammatory markers.

## Figures and Tables

**Figure 1 nutrients-17-00987-f001:**
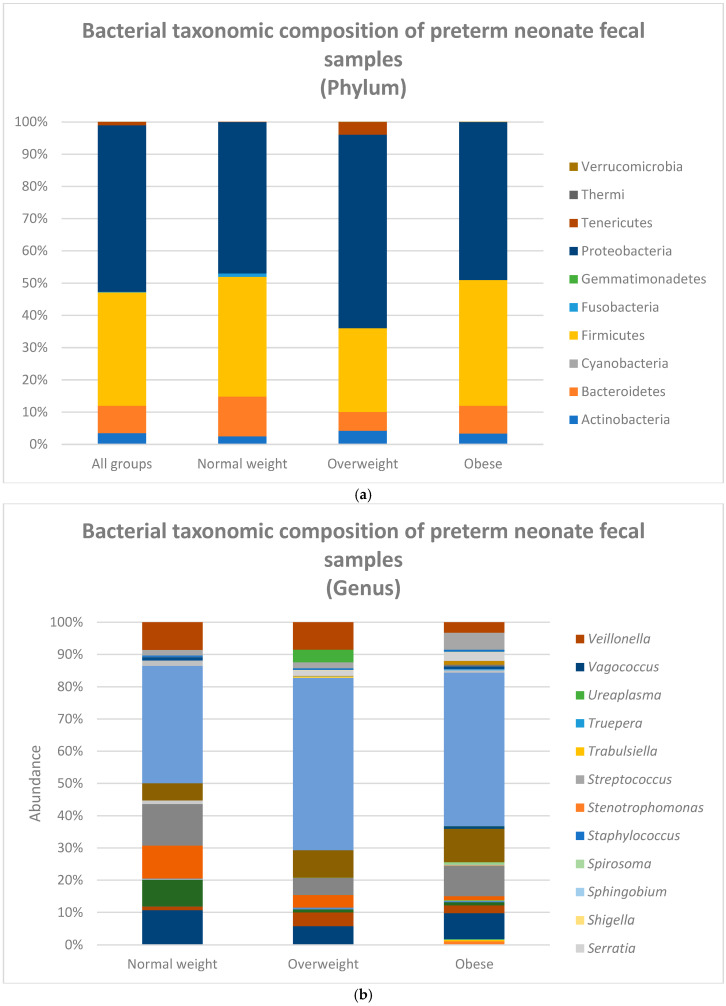
Relative abundances of microbiomes of premature neonates grouped by maternal pre-pregnancy BMI microbial population at the (**a**) phylum and (**b**) Genus levels for preterm infants grouped according to BMI of normal-weight (n = 10), overweight (n = 15), and obese (n = 29) mothers.

**Figure 2 nutrients-17-00987-f002:**
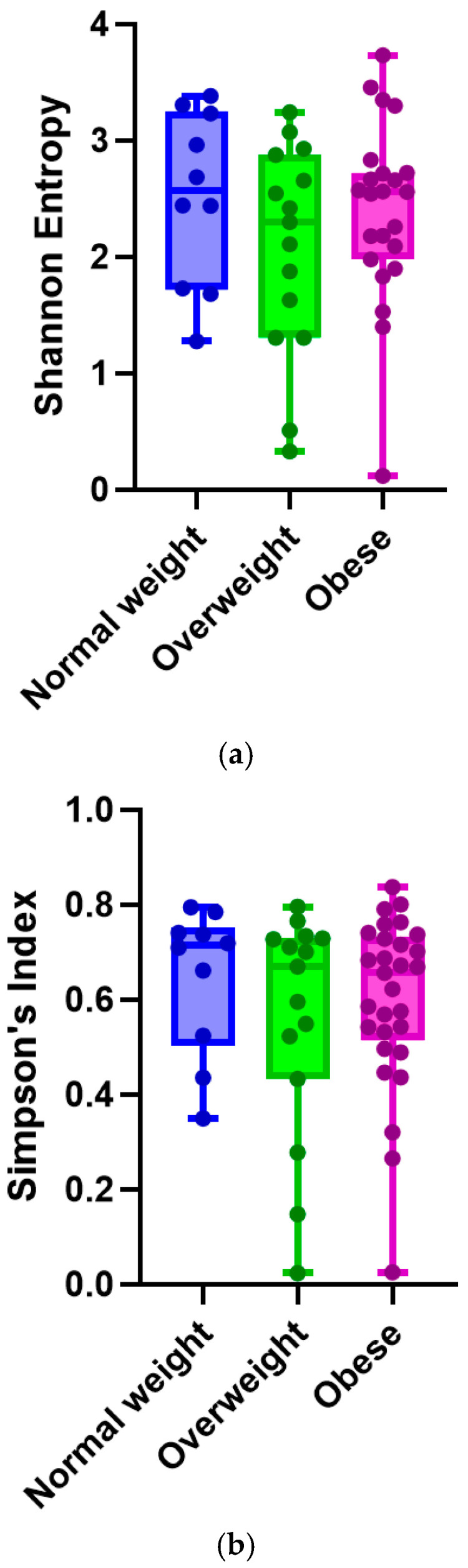
Alpha diversity based on (**a**) Shannon Entropy and (**b**) Simpson’s Index.

**Figure 3 nutrients-17-00987-f003:**
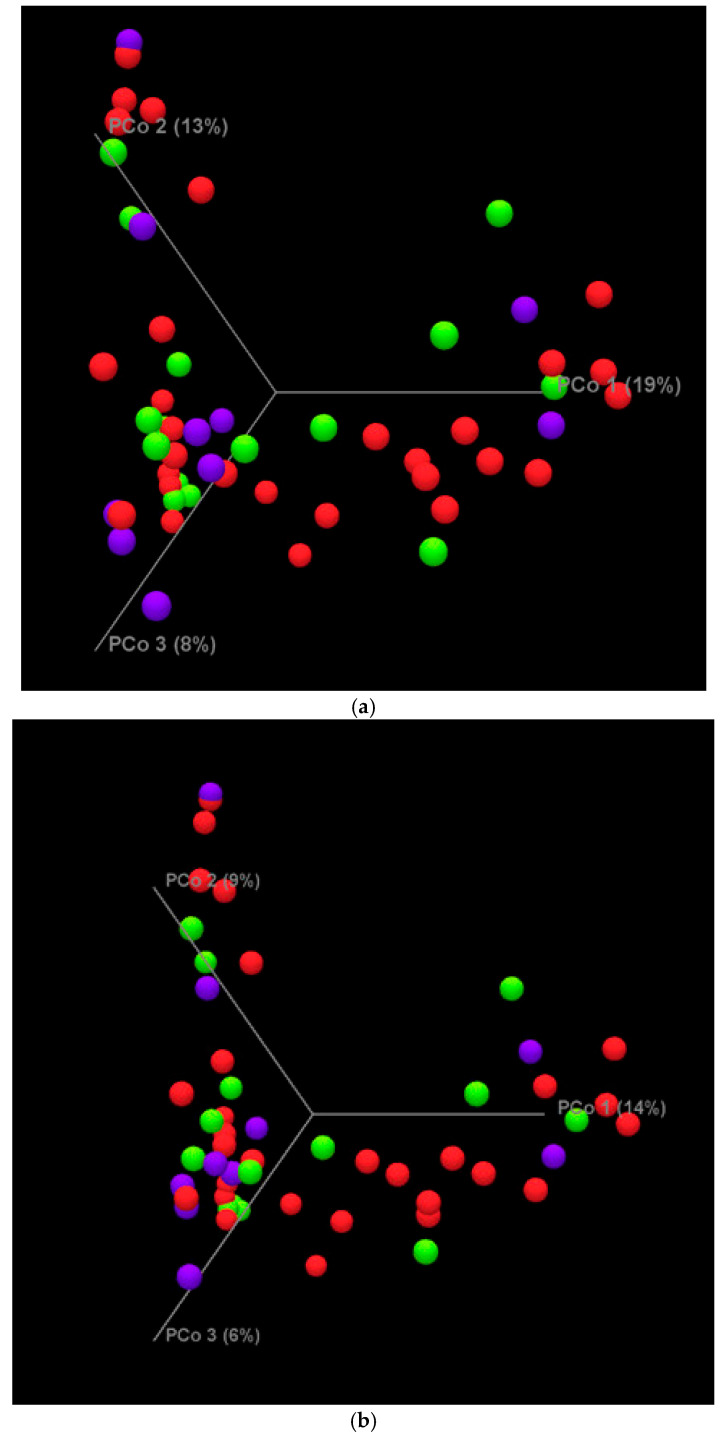
Beta diversity in (**a**) Bray–Curtis and (**b**) Jaccard PCoA plots of premature neonates grouped by maternal pre-pregnancy BMI. The groups are of infants born to normal-weight mothers (n = 10) in purple, overweight (n = 15) in green, and obese (n = 29) in red.

**Figure 4 nutrients-17-00987-f004:**
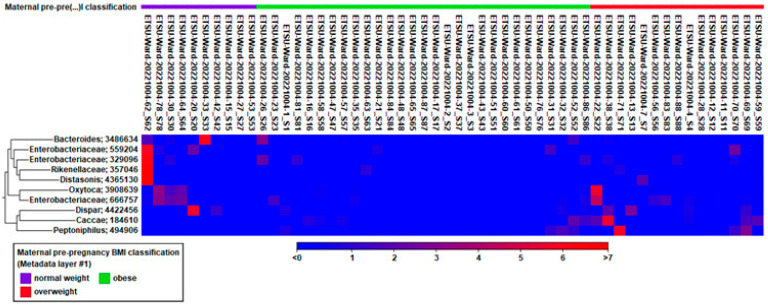
Differential abundances of taxa of premature neonates grouped by maternal pre-pregnancy BMI.

**Table 1 nutrients-17-00987-t001:** Maternal and Premature Infant Factors ^1^.

	Min–Max	Mean (SD)	N (%)
**Maternal Factor**			
**Pre-pregnancy BMI**	18–46.7	30.75 (6.92)	
** Normal weight**			10 (19%)
** Overweight**			15 (27%)
** Obese**			29 (54%)
**Race/ethnicity**			
** Black**			25 (46%)
** White**			20 (37%)
** Hispanic**			7 (13%)
** Other**			2 (4%)
**Maternal diabetes**			
** No**			39 (72%)
** Yes**			15 (28%)
**Infant Factor**			
**Gestational age**	169–242 days	208 (19.09)	
** Extremely preterm** (**22–27 weeks**)			14 (26%)
** Very preterm** (**28–31 weeks**)			25 (46%)
** Moderately preterm** (**32–37 weeks**)			15 (28%)
**Birth Weight**	620 g–2350 g	1263 (504.41)	
** Micro-preemie** (**<800 g**)			9 (16.5%)
** ELBW** (**<1000 g**)			9 (16.5%)
** VLBW** (**<1500 g**)			15 (28%)
** LBW** (**<2500 g**)			21 (39%)
** NBW** (**>2500 g**)			0 (0%)
**Fenton Growth %**			
** Birth weight**	0.1–93.3%	32.94 (26.63)	
** SGA** (**<10%**)			5 (9%)
** AGA** (**10–90%**)			48 (89%)
** LGA** (**>90%**)			1 (2%)
** Birth length**	0.2–99.1%	29.54 (31.61)	
** SGA** (**<10%**)			7 (13%)
** AGA** (**10–90%**)			39 (72%)
** LGA** (**>90%**)			8 (15%)
** OFC (occipital frontal circumference)**	0.2–92%	23.53 (23.99)	
** SGA** (**<10%**)			9 (17%)
** AGA** (**10–90%**)			44 (81%)
** LGA** (**>90%**)			1 (2%)
**Delivery Mode**			
** Vaginal**			15 (28%)
** Cesarean**			39 (72%)
**Fecal sample collection**	2–95 days	15.63 (19.24)	
** Week 1**			9 (17%)
** Week 2**			15 (28%)
** Week 3**			15 (28%)
** Week 4**			5 (9%)
** Week 5 +**			10 (18%)
**Feeding milk type**			
** Mom only**			5 (9%)
** Donor only**			5 (9%)
** Combination**			44 (81%)

^1^ Definitions for abbreviations listed in Table 1: BMI = Body Mass Index, ELBW = Extremely Low Birth Weight, VLBW = Very Low Birth Weight, LBW = Low Birth Weight, NBW = Normal Birth Weight, SGA = Small for Gestational Age, AGA = Average for Gestational Age, and LGA = Large for Gestational Age.

**Table 2 nutrients-17-00987-t002:** Premature infant characteristics by maternal pre-pregnancy BMI (kg/m^2^).

	Gestation (Days)	Birth Weight (g)	Fenton Growth (Birth Weight)	Fenton Growth (Birth Length)	Fenton Growth (Birth OFC)	Day of Life at Sample Collection	Race ETHNICITY (BLACK)	Maternal Diabetes (Yes)	C-Section	Feeding Type (Mom Only Milk)
**Normal BMI** **(<25 kg/m^2^)** **(n = 10)**	216.02 (14.27)	1408.15 (429.56)	35.44 (16.76)	37.80 (20.54)	22.96 (15.07)	12.72 (9.37)	3 (30%)	2 (20%)	6 (60%)	1 (10%)
**Overweight BMI** **(25–29.9 kg/m^2^)** **(n = 15)**	209.86 (20.28)	1346.17 (549.69)	43.53 (22.93)	50.17 (31.37)	32.11 (23.28)	24.10 (22.07)	7 (33%)	3 (14%)	13 (62%)	5 (25%)
**Obese BMI** **(≥30 kg/m^2^)** **(n = 29)**	207.49 (19.24)	1330.59 (510.11)	49.32 (28.98)	51.24 (31.47)	39.40 (26.04)	20.59 (19.30)	22 (55%)	14 (35%)	27 (66%)	4 (10%)
** *p* ** **-value**	0.3	0.3	0.3	0.7	0.2	0.5	0.3	0.2	0.9	0.4

Continuous data reported as mean (SD). Categorical data reported as n (%).

**Table 3 nutrients-17-00987-t003:** Multivariable linear regression models include covariates of race/ethnicity, c-section, infant day of life, gestational age, pre-pregnancy BMI, and % of MoM dose.

Model		Beta	SE	*p*-Value
**Acetic acid ***	Pre-pregnancy BMI one unit increase	−0.5137	0.257	0.051
**Butyric acid ***	Pre-pregnancy BMI one unit increase	0.02122	0.069	0.76
**Propionic acid ***	Pre-pregnancy BMI one unit increase	0.5914	0.248	**0.021**
**APB acids combined ***	Pre-pregnancy BMI one unit increase	0.09892	0.126	0.44
**Shannon diversity index (alpha diversity) ***	Pre-pregnancy BMI one unit increase	0.004639	0.014	0.74

* All models are adjusted for four covariates expected to be related to microbiome: race/ethnicity, c-section at birth, day of life of sample, gestational age, pre-pregnancy BMI, and % of MoM dose. Bold indicates a significant *p*-value.

**Table 4 nutrients-17-00987-t004:** Microbiome Results (*p*-values) from Factors other than Maternal Pre-pregnancy BMI.

	Alpha Diversity		Beta Diversity	
	Kruskal–Wallis	Shannon Diversity	Bray–Curtis	Jaccard
**Infant weeks of life**	0.1	0.55	0.14	0.20
**Gestational age**	0.9	0.46	>0.05	0.06
**Delivery mode**	0.2	0.79	**0.02**	**0.03**
**Race/ethnicity**	0.3	0.30	0.99	0.99
**Feeding type**	0.1	0.63	0.48	0.50
**MoM dose**	0.1	0.74	0.66	0.73

Bold indicates a significant *p*-value.

## Data Availability

The original contributions presented in the study are included in the article, further inquiries can be directed to the corresponding author.

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
