# Peer review of "Effects of Maternal Pre-Pregnancy BMI on Preterm Infant Microbiome and Fecal Fermentation Profile—A Preliminary Cohort Study"

_nutrients, 2025, doi:10.3390/nu17060987_

Round 1

Reviewer 1 Report

Comments and Suggestions for Authors

This is a preliminary cohort study with 53 enrollees over ten months. Microbiomes from neonatal stool were investigated, and the methodology is well-described. Unfortunately, I personally feel the study might provide more information if fecal microbiomes from the corresponding mothers were also obtained. This is a good start for such a study, which has the potential to help us understand mother’s anthropometric impact on the long-term health of their offspring. I also do not feel comfortable about using the word “adiposity” in your topic as BMI does not measure adiposity directly, although I have to agree that the two terminologies are highly correlated (r=0.63, p<0.01 in Rai R, et al. Relationship Between Body Mass Index and Body Fat Percentage in a Group of Indian Participants: A Cross-Sectional Study From a Tertiary Care Hospital. Cureus. 2023 Oct 27;15(10):e47817).

It is probably true that in animal studies, the offspring of obese mothers are more susceptible to obesity. However, only “male” offspring showed this relationship in the cited reference. Interestingly, the Dutch Famine Birth Cohort (DFBC) research showed an opposite relationship that linked prenatal famine exposure to increased risk of obesity later in life. The offspring of both obese mothers and nutrition-deprived mothers are at risk for obesity.

Is there a good reason why 341F and 785R primers were used for this study, while 515F and 806R seem to have better coverage [Wasimuddin, et al. Evaluation of primer pairs for microbiome profiling from soils to humans within the One Health framework. Mol Ecol Resour. 2020 Nov;20(6):1558-1571]? The 341F/785R primers are claimed to be more suitable for older individuals with well-established bacterial communities [Palkova L, et al. Evaluation of 16S rRNA primer sets for characterisation of microbiota in paediatric patients with autism spectrum disorder. Sci Rep. 2021 Mar 24;11(1):6781]. Can you provide the rationale for choosing the 341F/785R primer pair?

There was no alpha-diversity between different maternal BMI groups but significantly different beta-diversity microbiomes according to the mode of delivery (Table 4). This led to my question about the description from lines 272 to 286. Obese mothers frequently end up with cesarean sections, and antibiotics are thus frequently on antibiotics. Could it be possible that the findings are secondary to the mode of delivery? I ask this question because data were not provided like in section 4.3..

I do not agree with some parts of the discussion, such as the results showing no alpha- and beta-diversity between different BPI groups, but you still expect to see some difference. If your result does not support your assumption, then you should not claim it. It will be OK to argue that your small sample size leads to no significant difference, and more extensive study would be needed to verify your hypothesis. I also disagree with the viewpoint in lines 363-368 as the finding in children cannot directly explain the maternal BMI. This is just like butyrate is assumed to be beneficial but contrary findings were reported in adults.  

The only significant difference in diversity is the beta-diversity according to delivery mode. Interestingly, this significant finding was not discussed at all.  There are four paragraphs in the conclusion section. I will suggest moving most of the content to the discussion section. Avoid a lengthy conclusion, as it loses its purpose as a conclusion.

  • Lines 60-65: Please switch “MoM milk” (line 61) and “Mother's milk (MoM).”
  • Lines 97-98: Mother’s Own Milk (MoM) can use MoM.
  • Page 10: Why does Figure 2 have figure legend 3, and no figure legend of Figure 3 (page 11)?
  • Page 12: I want to make sure there is no figure legend 4.
  • Z
  • Line 276: … were all “significantly” different in fecal samples… Is it statistically significant or subjectively significant?

Author Response

We have provided a response to your comments and suggestions in the attached document. Our responses are in bold. Again, thank you very much for your time, we greatly appreciate you.

Reviewer 2 Report

Comments and Suggestions for Authors

This is an extremely interesting study that is well designed and the results are presented with an appropriate number of figures and tables.
However, the axes in the figures are illegible (supplement 1a-1d) and I also suggest that the labels in the figure description and in the figure itself be standardized (upper and lower case letters are combined, and it will be clearer if this is uniform)
Figures 1 and S1 - the more logical order is normal BMI, overweight and then obesity.

Author Response

(The authors gave the same response as above.)

Reviewer 3 Report

Comments and Suggestions for Authors

The article addresses a specific gap in the field related to how maternal adiposity (BMI) influences preterm infant microbiome development and fecal fermentation profiles. This study is among the first to analyze both microbiome composition and short-chain fatty acid (SCFA) levels in preterm infants while considering maternal BMI as a factor.

  1. The study was underpowered for microbiome analysis. A larger cohort (beyond 54 preterm infants) would provide greater statistical power to detect differences in microbiome diversity and SCFA levels.
  2. Infant fecal samples were collected between 2 and 95 days post-birth, introducing potential confounding due to microbiome changes with age. Limiting collection to a narrower window (30-45 days) would reduce this variability.
  3. The study adjusted for race and ethnicity, c-section, gestational age, and MoM dose, but additional factors such as maternal diet, medications, antibiotic exposure, and infant feeding initiation should be considered.

By addressing these aspects, the authors can improve the methodological rigor of their study and enhance the clinical applicability of their deep learning models.

Author Response

(The authors gave the same response as above.)

Round 2

Reviewer 1 Report

Comments and Suggestions for Authors

I am comfortable with your response and have no more comment.

Author Response

Reviewer, thank you very much for your time, we greatly appreciate you.